# Prompting Strategies for Enterprise Text-to-SQL: An Ablation Study

Omar Bouattour
TU Munich and MIT
omarboua@mit.edu

Joyce Weiner*
Intel
joyceweiner@gmail.com

Nesime Tatbul
Intel and MIT
tatbul@csail.mit.edu

Fabian Wenz
TU Darmstadt and MIT
fab_wenz@mit.edu

Peter Baile Chen
MIT
peterbc@mit.edu

Layne Mills†
MIT
layne.mills@gmail.com

Yong He
Intel
yong.m.he@intel.com

Bijan Arbab
UCSD
bijanarbab@gmail.com

Çağatay Demiralp
AWS AI Labs and MIT
cagatay@csail.mit.edu

Michael Stonebraker
MIT
stonebraker@csail.mit.edu

## ABSTRACT

Recent advances in Large Language Models (LLMs) have improved Text-to-SQL generation, yet enterprise-scale databases remain challenging due to large schemas, domain-specific terminology, and noisy retrieval conditions. We present a modular text-to-SQL pipeline for a real Intel telemetry use case and conduct an ablation study of prompt augmentation, schema annotations, few-shot examples, and iterative SQL validation under varying schema exposure settings. Beyond execution accuracy, we evaluate robustness, execution failures, latency, token consumption, and inference cost. Experiments with GPT-4.1 show that effective enterprise text-to-SQL systems require making careful tradeoffs among schema retrieval, prompt design, and operational cost.

**VLDB Workshop Reference Format:**
Omar Bouattour, Joyce Weiner, Nesime Tatbul, Fabian Wenz, Peter Baile Chen, Layne Mills, Yong He, Bijan Arbab, Çağatay Demiralp, and Michael Stonebraker. Prompting Strategies for Enterprise Text-to-SQL: An Ablation Study. VLDB 2026 Workshop: Novel Optimizations for Visionary AI Systems (NOVAS).

## 1 INTRODUCTION

Text-to-SQL systems translate natural language questions into executable SQL queries. Recent advances in Large Language Models (LLMs) have substantially improved their ability to generalize across databases and generate complex SQL programs [6]. Models such as GPT-4 demonstrate strong zero-shot and few-shot reasoning capabilities, making natural language interfaces to databases increasingly practical.

Despite this progress, enterprise text-to-SQL remains considerably more challenging than traditional academic benchmarks [1]. Enterprise databases often contain hundreds or thousands of tables, extensive metadata, noisy naming conventions, and specialized domain terminology [8]. In such environments, providing the full schema frequently exceeds context limits and introduces reasoning noise, requiring systems to balance schema retrieval, prompt engineering, validation mechanisms, and deployment cost.

Most existing evaluations focus primarily on execution accuracy. However, practical deployments must also consider latency, monetary cost, robustness, and retrieval-induced errors. Consequently, enterprise text-to-SQL should be evaluated not only for correctness, but also for efficiency, robustness, and operational tradeoffs.

This work presents a modular text-to-SQL framework for systematically studying these factors. The framework supports configurable prompt construction, schema retrieval, validation, and execution-based evaluation. In particular, the study explores three prompt augmentation components: (i) *Examples*: few-shot demonstrations consisting of natural language questions paired with SQL queries. (ii) *Annotations*: natural language schema descriptions and semantic metadata. (iii) *Validation*: iterative SQL verification and corrective regeneration. The modular structure of our pipeline enables evaluating these components individually as well as in all possible combinations, under multiple schema exposure settings, including oracle gold-table access, retrieval-based schema selection, and full-schema prompting.

Our work is motivated by a real-world enterprise use case from industry. More specifically, we used a representative data sample from Intel's large-scale telemetry dataset as well as a set of SQL queries from real query logs, for which domain experts from Intel produced natural-language descriptions, to run a detailed ablation study over our modular pipeline.

---

*Work completed while at Intel.

†Work completed while at MIT.

Proceedings of the VLDB Endowment. ISSN 2150-8097.

Our experimental analysis reveals several important observations about real-world enterprise text-to-SQL applications: (i) Schema annotations, when available, consistently provide the strongest individual gains across nearly all retrieval settings. (ii) Few-shot examples become substantially more effective when combined with annotations, suggesting strong complementarity between semantic grounding and structural demonstrations. (iii) Validation primarily improves robustness by reducing execution failures rather than dramatically increasing semantic correctness. (iv) Finally, retrieval quality strongly affects prompt effectiveness, highlighting important tradeoffs between retrieval depth, prompt complexity, and inference efficiency.

From a business perspective, these results reinforce the benefits of having good technical documentation, since schema annotation context is supported by having high-quality descriptions for database tables and columns. Indeed, the detailed data dictionary document that was available for the Intel telemetry dataset was influential in boosting SQL generation performance in our pipeline.

Our modular text-to-SQL framework is applicable to a broad range of enterprise use cases. We believe that our ablation study along with the analysis of deployment tradeoffs can provide insights into making data-supported business decisions when creating text-to-SQL solutions, balancing effectiveness against cost and efficiency.

The contributions of this work can be summarized as follows:

- We present a modular framework for enterprise text-to-SQL, supporting table schema retrieval, prompt augmentation, validation, and execution-based evaluation.
- We conduct a comprehensive ablation study of text-to-SQL prompting strategies based on this modular framework and a real-world use case from Intel, evaluating the interaction between examples, annotations, and validation under multiple schema exposure settings.
- We analyze practical deployment tradeoffs involving accuracy, robustness, token consumption, latency, and monetary cost.
- We discuss key takeaways from our industry case study for informing the future development of more robust enterprise text-to-SQL solutions.

## 2 RELATED WORK

Others have also observed the challenges of building LLM-based text-to-SQL capabilities in enterprise settings, proposing new techniques to quantify and address these challenges. Recent evaluation research includes the BEAVER benchmark based on real enterprise data/query workloads and fine-grained annotations [1], and Bench-Press – a system to create similar custom benchmarks in any enterprise [8]. Gurajada et al. study prompt optimization techniques for text-to-SQL systems, focusing on the exemplar selection problem [3]. Eckmann et al. propose a human-like iterative approach for LLM-based SQL generation of complex queries, focusing on those with a high number of join clauses [2]. Li et al. propose a software-engineering-inspired framework for text-to-SQL generation. Their approach incorporates software engineering unit-testing principles to verify the correctness of generated SQL queries and provide stronger correctness guarantees [4]. Our work builds on and expands these efforts by providing a detailed experimental study of

a modular enterprise text-to-SQL pipeline based on an industry case study, revealing a number of new observations and insights. Mohammadreza Pourreza et al. performed a similar ablation study to evaluate the effect of task decomposition on text-to-SQL accuracy [7]. They also introduced an LLM-based self-correction step to detect and fix erroneous SQL, whereas our validation is algorithmic and rule-based. While their experiments were conducted on public benchmarks, ours target enterprise Intel data. Despite these differences, both studies reach similar conclusions.

## 3 ENTERPRISE TEXT-TO-SQL USE CASE

Our enterprise use case comes from Intel's telemetry data collection and analytics team. Intel collects petabytes of telemetry data from millions of end-user devices deployed worldwide, each of which sends data on a daily basis. More than 1000 metrics are collected from each opt-in client device in a privacy-preserving manner. The goal is to gain insights into the real-world usage of hardware components, identify potential issues with system performance and user experience, and drive future business decisions.

The dataset contains a wide variety of information about system configuration, including hardware and software characteristics such as CPU, memory, storage, and operating system related information. In addition to such static data, the collector also records dynamic usage and state metrics, including application activity, process behavior, power states, thermal conditions, device usage, and network performance. This combination of device configuration and temporal telemetry data results in complex table schemas with heterogeneous data types and rich inter-table relationships. Furthermore, the data collection system is organized across multiple processing layers that reflect the lifecycle of telemetry data. At the collector level, raw system metrics are gathered locally and stored on the host system. These data are subsequently processed and transformed through intermediate analysis stages before being integrated into large-scale analytical storage systems. This multi-stage processing pipeline results in schemas that evolve over time and include both low-level measurements and derived metrics.

This rich dataset is of interest to multiple different types of users at Intel, including engineers, data scientists, and business decision-makers. Hence, text-to-SQL is a key capability to make the data more accessible to this wide range of users.

Compared to the datasets of commonly used public text-to-SQL benchmarks [5, 9], the Intel dataset exhibits greater schema complexity, domain-specific terminology, and temporal dynamics. The presence of evolving schemas and specialized telemetry signals introduces additional challenges for schema grounding and query generation, as is often the case for other real enterprise scenarios [1].

In this research, we worked with a representative sample of the Intel telemetry dataset, which consists of 25 tables (out of more than 100 in the original dataset). On average, there are around 500K rows and 16 columns per table in our smaller sample, which is already larger in scale than all existing benchmarks [1, 5, 9].

Our study uses 30 natural-language queries, which were derived from logs of SQL queries written by Intel domain experts as part of real analytical workflows. Domain experts subsequently produced the corresponding natural-language descriptions , forming

realistic NL-SQL pairs that reflect genuine information needs. Compared to the other benchmarks, our Intel queries are substantially longer, with more keywords (around 25 on average compared to 16 in Beaver [1]) and tokens (around 140 on average compared to 100 in Beaver [1]), indicating more verbose and detailed analytical requests typical of real operational workflows. The workload also exhibits a markedly higher number of aggregation operations (14 aggregations on average compared to 6 in Beaver [1]), suggesting that telemetry analysis is heavily driven by summary statistics, normalization, and derived metrics. In contrast, Intel queries reference fewer tables on average than Beaver (1.67 vs. 4.2), reflecting a pattern where analysis is often performed on large, wide fact tables rather than through extensive joins. The nesting depth (1.63 on average) remains comparable to other benchmarks, indicating moderate use of subqueries and layered computations.

In general, our Intel telemetry workload represents a distinct complexity profile characterized by long analytical queries, aggregation-heavy computations, and large-scale telemetry data. These properties differentiate it from academic benchmarks [5, 9] and align it more closely with workloads from real enterprise analytical environments [1].

## 4  A MODULAR TEXT-TO-SQL PIPELINE

We designed a modular text-to-SQL pipeline, as illustrated in Figure 1. Given a natural language question (1), we first provide a schema context using one of three options (2): retrieval-based schema selection (top-k most relevant tables), full database schema provision (all tables in the DB), or gold schema provision (all tables needed by the gold SQL). The selected context can then be enriched with optional prompt augmentation components (3), namely, few-shot examples (3a) and/or schema annotations (3b). The LLM generates an initial SQL query (4), which may optionally pass through an iterative validation stage that checks SQL correctness and provides feedback for regeneration when errors are detected (5). The final SQL query is executed against the database (6) and its result is compared with the result of the gold SQL query using execution-based evaluation (7), leading to an execution accuracy score (8). We now describe each module in more detail:

**Table Selection.** The table selection module allows multiple ways to select table schemas:

- Retrieved Tables: The model receives only a subset of the most relevant table schemas (top-$k$ tables) to be selected by a retriever. The $k$ parameter can be tuned considering the complexity of the query and the desired context size. In our implementation, we use a hybrid dense/sparse schema retriever designed specifically for enterprise telemetry schemas as in our use case. The retriever combines semantic dense retrieval using the BAAI bge-base-en-v1.5 embedding model with sparse BM25 lexical retrieval. Table representations are constructed using: table names, database identifiers, table descriptions, column names, column types, and column descriptions. Dense retrieval computes cosine similarity between question embeddings and schema embeddings. Sparse retrieval computes BM25 relevance scores over textual schema representations. For hybrid retrieval, both dense and sparse scores are independently normalized on a per-query basis and combined through score summation. Candidate tables are then ranked and the top-$k$ highest-scoring tables are selected for prompt construction. This hybrid design was chosen to balance semantic matching and exact lexical overlap, particularly for technical schema terminology and telemetry-specific column names as in our use case.

- All Tables: The model simply receives the full DB schema of all of the tables in the database. While this option is easy to set up for any use case, it may increase the required context size for large enterprise databases with many tables. Furthermore, in use cases with extremely verbose schemas like ours, this setting could also help test the model's robustness to noisy and irrelevant context.

- Gold Tables: The model receives only the schemas of the gold tables required to answer each question. Although it may be unrealistic to apply in complex enterprise settings where gold tables are not precisely known, this option removes schema retrieval as a source of error and approximates an oracle upper bound for SQL generation quality.

**Context Enrichment.** The context enrichment module provides two components that can be optionally enabled:

- Few-Shot Examples: This component adds in-context demonstrations consisting of natural language questions paired with SQL queries. These demonstrations are intended to guide the model toward appropriate SQL structures, aggregation patterns, and formatting conventions. The examples are selected statically and inserted directly into the system prompt prior to the target question.

- Schema Annotation: This component enriches schema representations using natural-language descriptions of tables and columns. For our use case, these annotations provide semantic explanations for telemetry metrics, system measurements, and domain-specific terminology from a data dictionary document provided by Intel. The goal of annotations is to improve schema understanding and reduce ambiguity during SQL generation.

**SQL Validation.** The validation component performs iterative verification of the SQL generated by the LLM. Unlike most related work, which relies on an LLM for self-correction, our validation is entirely algorithmic and rule-based. Generated SQL queries are checked for formatting violations, malformed outputs, missing answer tags, and SQL structural inconsistencies. If the validation fails, the model receives corrective feedback describing the issue and is prompted to regenerate a corrected SQL query. This process may repeat for a predefined number of retries. In practice, a single regeneration was sufficient in most cases to transform a non-runnable SQL query into a runnable one. Validation therefore acts as a lightweight self-correction mechanism without requiring external symbolic repair systems.

## 5  ABLATION STUDY

To analyze the impact of various modules in our text-to-SQL pipeline on execution accuracy, we conducted an ablation study. In this section, we describe our experimental setup and results.

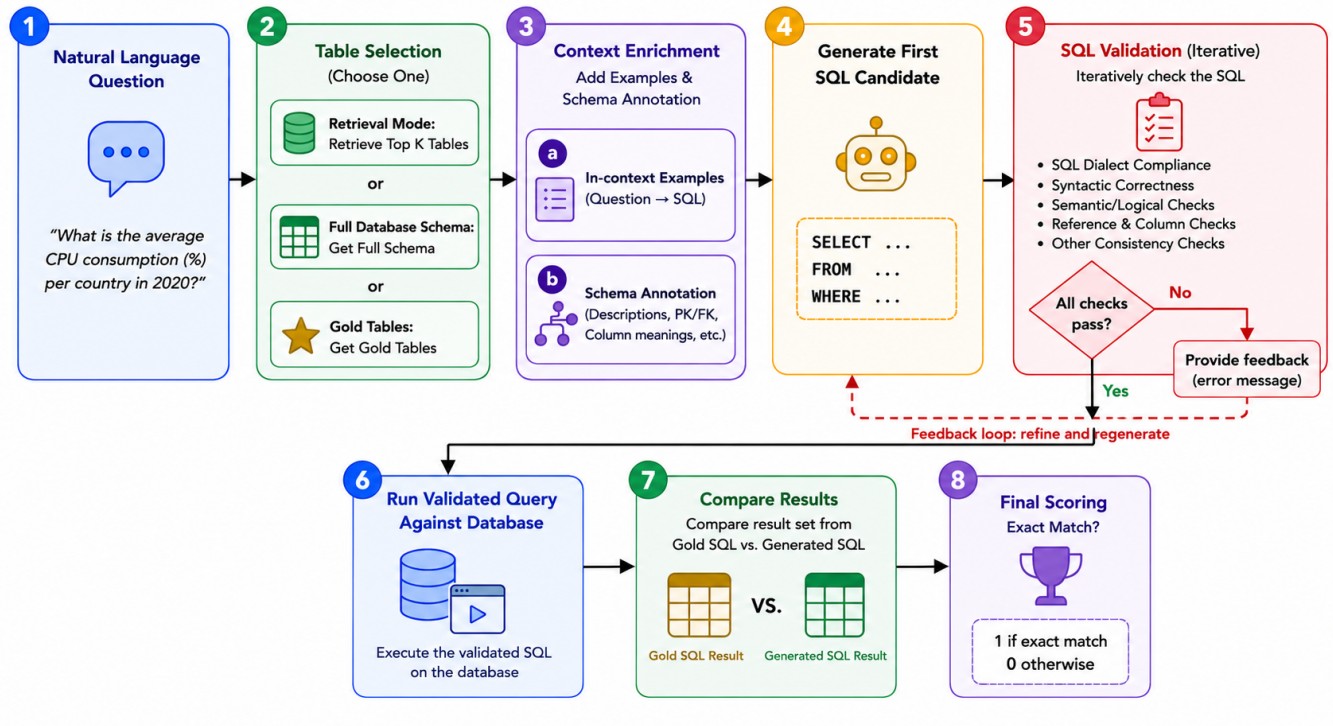

Figure 1: Overview of the modular text-to-SQL pipeline.

## 5.1 Setup

**Pipeline Configurations.** We evaluate all eight possible combinations of the context enrichment and SQL validation components: (i) Base (LLM-only), (ii) Examples, (iii) Annotations, (iv) Validation, (v) Examples + Validation, (vi) Annotations + Validation, (vii) Examples + Annotations, (viii) Examples + Annotations + Validation. Furthermore, each pipeline configuration is evaluated under multiple table selection settings, resulting in a comprehensive ablation study of our modular text-to-SQL pipeline.

**Model Configuration.** All experiments use GPT-4.1 as the underlying SQL generation model. To ensure deterministic behavior and reproducibility, temperature is fixed to zero for all runs. The maximum generation length is set to 4096 tokens. The same model configuration is used across all experiments to isolate the effects of prompt engineering and schema retrieval from variability introduced by model sampling or decoding strategies.

**Dataset.** All experiments are conducted with data and queries from a representative subset of our enterprise use case. This includes 25 tables and 30 manually curated natural language questions paired with gold SQL queries. Together, these questions span a wide range of SQL patterns and reasoning requirements, including: multi-table joins, aggregations and grouping, temporal filtering, ranking and ordering, statistical summarization, system telemetry analysis, and schema interpretation.

**Evaluation Metrics.** The primary evaluation metric is execution accuracy. A generated SQL query is considered correct if executing the query produces the same output as the corresponding gold SQL query (a score of 1), and incorrect otherwise (a score of 0).

Execution accuracy evaluates query result equivalence rather than syntactic similarity, making it more appropriate for enterprise analytical workloads where multiple alternative SQL formulations may produce identical outputs.

Our comparative evaluator supports several equivalence relaxation mechanisms to avoid penalizing minor output differences, including: order-invariant result comparison, column alignment correction, numeric tolerance matching, unordered multiset comparison. These mechanisms improve evaluation stability for analytical SQL workloads involving aggregation, minor floating-point computations, and unordered query outputs.

For failed executions (non-runnable queries), we also keep track of error types, including: malformed SQL syntax, invalid table references, invalid column references, SQL dialect violations, runtime execution failures.

Finally, beyond accuracy, we also record operational performance and cost metrics, including: number of input/output/total tokens, end-to-end SQL generation latency, number of validation retries, and monetary cost of model inference.

## 5.2 Results

**Execution Accuracy (Gold Tables).** Figure 2 presents accuracy results for the Gold-Tables setting, where the model receives exactly the relevant tables required to answer each query. This setting removes the influence of schema retrieval, allowing the effects of the remaining pipeline components to be evaluated in isolation. Several important observations emerge from these results: (i) First, the Base configuration performs relatively poorly, achieving only 43.3%

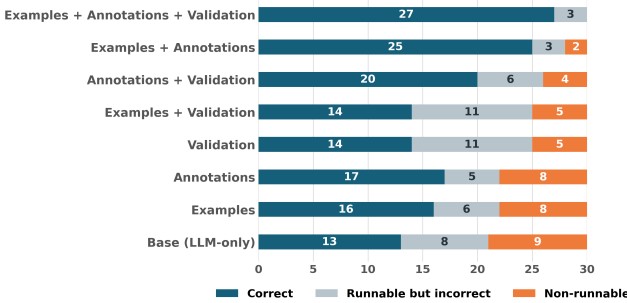

**Figure 2: Execution Accuracy (Gold Tables).**

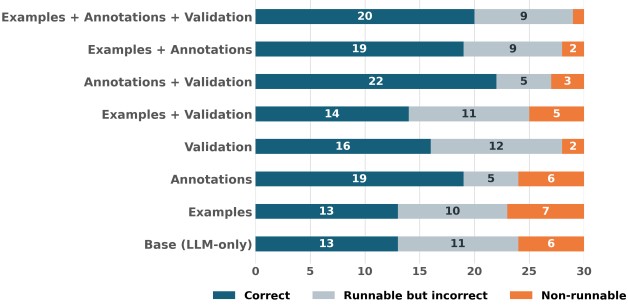

**Figure 3: Execution Accuracy (Top-10 Table Retrieval).**

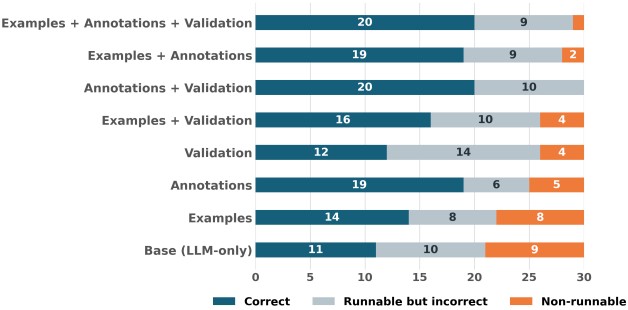

**Figure 4: Execution Accuracy (All Tables).**

execution accuracy. This demonstrates that enterprise telemetry schemas remain difficult even when the relevant tables are known exactly. (ii) Second, Validation alone primarily improves robustness rather than semantic correctness. While execution errors decrease substantially from 9 to 5, overall accuracy improves only marginally. This suggests that many failures in the Base configuration are semantic rather than syntactic. (iii) Third, Annotations provide the strongest individual gains among all isolated components. Adding semantic schema explanations increases accuracy from 43.3% to 56.7%, indicating that schema interpretation is one of the dominant bottlenecks in enterprise text-to-SQL. (iv) Fourth, the strongest improvement occurs when Examples and Annotations are combined. This configuration achieves 83.3% accuracy, while reducing errors to only 2. This result suggests a strong complementarity between semantic grounding and structural demonstrations. Examples help the model learn SQL structure and reasoning patterns, while annotations improve understanding of telemetry terminology and schema semantics. (v) Finally, the complete configuration combining Examples, Annotations, and Validation achieves the best overall result with 90.0% accuracy and zero execution errors. This indicates that Validation contributes additional robustness once semantic grounding and structural guidance are already strong.

**Execution Accuracy (Top-10 Table Retrieval).** Figure 3 reports results for the Top-10 Table Retrieval setting, where the model receives the ten most relevant tables returned by the retriever. Compared to the Gold Table setting, the Table Retrieval setting introduces substantial additional difficulty. Although the top-10 retriever generally provides relevant schema, the retrieved context occasionally includes noisy or partially irrelevant tables, increasing the complexity. Interestingly, Examples alone no longer improve over the Base configuration. This suggests that few-shot demonstrations become less effective when schema retrieval introduces noise or ambiguity. The model may struggle to generalize demonstrations under partially incorrect schema contexts. Annotations, however, remain highly effective. Accuracy increases from 43.3% to 63.3% when Annotations are added, confirming that semantic schema grounding remains valuable even under imperfect retrieval conditions. The best-performing configuration in this setting is Validation combined with Annotations, achieving 73.3% accuracy. Validation becomes more important under retrieval noise, because retrieval errors increase the likelihood of malformed SQL or invalid

references. Unexpectedly, the full configuration slightly underperforms Validation with Annotations. One possible explanation is that adding Examples increases prompt complexity and token length, which may interfere with reasoning when the retrieved schema already contains noise.

**Execution Accuracy (All Tables).** Figure 4 presents results for the All Tables setting, where the model receives the complete database schema. The All Tables setting is the most challenging configuration overall. Providing the entire enterprise schema introduces substantial context noise and increases the difficulty of identifying relevant tables and columns. The Base configuration performs particularly poorly, achieving only 36.7% accuracy. This demonstrates that large-context prompting alone is insufficient for enterprise telemetry databases. Annotations again provide the largest individual gains, improving performance by more than 26 percentage points. This confirms that semantic grounding becomes increasingly important as schema complexity grows. Validation significantly reduces execution errors in this setting, especially when combined with Annotations. The Validation + Annotations configuration achieves zero execution failures, demonstrating strong robustness under extremely noisy schema conditions. Interestingly, the full configuration does not outperform Validation with Annotations, again suggesting that examples may introduce diminishing returns or additional prompt interference under large schema contexts.

**Retrieval Depth Analysis.** To better understand the interaction between retrieval depth and prompt augmentation, additional experiments were conducted using Top-3 and Top-5 retrieval settings.

*Top-3 Table Retrieval.* The Top-3 setting provides the smallest schema context and therefore tends to yield the lowest prompt size and inference cost. However, retrieval recall is also lower, increasing the probability that relevant tables are omitted entirely. The base configuration achieves only 26.7% accuracy in this setting. Performance improves substantially when annotations are introduced, and the best-performing configuration achieves 66.7% accuracy. These results indicate that aggressive schema compression can significantly reduce reasoning quality if retrieval recall becomes insufficient.

*Top-5 Table Retrieval.* The Top-5 setting provides a compromise between retrieval recall and prompt size. Compared to Top-3 retrieval, accuracy improves across nearly all configurations while maintaining substantially lower prompt complexity than Top-10 retrieval. The strongest configuration achieves 66.7% accuracy with only one execution error.

Overall, the retrieval depth experiments reveal a classical tradeoff between retrieval recall and prompt complexity. Increasing retrieval depth improves schema coverage, but also increases token consumption, latency, and context noise.

**Performance and Cost.** Beyond execution accuracy, enterprise text-to-SQL systems must also be evaluated in terms of operational deployment efficiency. Large prompts, iterative validation, and retrieval overhead can significantly increase inference latency, token consumption, and monetary cost. High execution accuracy alone is insufficient for evaluating the practicality of a text-to-SQL system in real-world enterprise deployments. To better understand these tradeoffs, we record several operational metrics for every SQL generation, including input tokens, output tokens, latency, validation retries, and estimated inference cost. Table 1 summarizes these measurements for the Gold Table setting experiments.

Several important observations emerge from these results: (i) First, configurations that achieve the strongest execution accuracy also incur the highest inference cost. The full configuration combining Examples, Annotations, and Validation achieves the highest execution accuracy, at 90.0%, but also produces the largest prompts, the highest latency, and monetary cost. This shows that increasing SQL generation quality often requires substantially more computation and context. (ii) Second, few-shot examples contribute heavily to prompt size growth. Because demonstrations include both natural language questions and complete SQL programs, they significantly increase the number of input tokens processed by the model. While Examples can improve structural reasoning, they also increase latency and inference cost. (iii) Third, Validation introduces an additional form of computational overhead. Failed SQL generations trigger regeneration cycles, increasing both total latency and token consumption. However, Validation also substantially reduces execution failures, particularly under noisy retrieval conditions. This highlights an important tradeoff between robustness and efficiency. (iv) Annotations also increase prompt size because enterprise telemetry schemas contain extremely verbose natural-language metadata. Nevertheless, annotations consistently provide strong accuracy gains relative to their additional cost, suggesting that semantic schema understanding is highly valuable for enterprise text-to-SQL systems. (v) Although the complete pipeline achieves the highest execution accuracy, the marginal gain analysis reveals that not all enhancements provide the same return on additional inference cost. Among the individual components,

schema annotations offer the largest improvement per unit cost, while combining examples with validation provides the smallest return despite increasing token usage and latency. This suggests that annotations are the most cost-effective standalone enhancement, whereas the full pipeline is preferable when maximizing execution accuracy outweighs computational cost.

These results demonstrate that enterprise text-to-SQL optimization is fundamentally a multi-objective problem. Practical systems must jointly optimize for metrics beyond execution accuracy, including: robustness to SQL failures, prompt size, retrieval quality, inference latency, monetary deployment cost. This tradeoff becomes more important in realistic enterprise deployments, where systems may process thousands of queries per day and inference costs accumulate rapidly. Thus, future systems should move towards adaptive strategies capable of dynamically balancing accuracy and efficiency, depending on query complexity and operational constraints. Such adaptivity could be achieved by selecting pipeline components based on query characteristics and available resources. For example, simple queries with high retrieval confidence could be processed using only the base pipeline, while more complex queries or those with lower retrieval confidence could trigger additional components such as schema annotations, few-shot examples, or iterative validation. This would allow the system to reserve computationally expensive operations for cases where they are expected to provide the greatest benefit, improving the trade-off between execution accuracy and inference cost.

## 6 DISCUSSION

The ablation study reveals several important insights into enterprise text-to-SQL generation and the interaction between prompt engineering and schema retrieval.

**The Importance of Schema Semantics.** Schema annotations consistently provide the largest performance improvements across retrieval settings. This effect is particularly strong in enterprise telemetry schemas, where abbreviated names and domain-specific terminology are difficult to interpret from raw schema information alone. Natural-language annotations provide semantic grounding that remains beneficial even under noisy retrieval and full-schema prompting, indicating that schema understanding is a major bottleneck in enterprise text-to-SQL.

**Few-Shot Examples Require Reliable Context.** Few-shot examples improve performance under Gold-table access, especially when combined with annotations, but provide limited benefit under noisy retrieval. This suggests that demonstrations are most effective when the retrieval system has high confidence in the selected tables (e.g., strong retrieval scores or clear agreement across retrieval signals). Since examples also increase prompt size and token consumption, they should be used selectively when retrieval confidence is high and sufficient context budget is available.

**Validation Primarily Improves Robustness.** Validation consistently reduces execution failures but yields only modest gains in semantic accuracy. While it can correct malformed SQL, it cannot fully recover from reasoning errors or missing schema understanding. Its value is greatest under noisy retrieval conditions, where iterative regeneration improves reliability. Validation should therefore be viewed primarily as a robustness mechanism.

**Table 1: Accuracy-performance-cost tradeoffs for the Gold-Table setting across pipeline configurations. Token and latency values are averaged per query. Relative changes in tokens, latency, and cost are measured with respect to the Base (LLM-only) configuration. Marginal Gain reports the increase in execution accuracy (percentage points) obtained per additional $0.01 of inference cost relative to the Base configuration.**

| Configuration | Accuracy | Non-runnables | Avg Input Tokens | Avg Output Tokens | Avg Latency (ms) | Avg Cost ($) | Marginal Gain (pp/$0.01) |
|---|---|---|---|---|---|---|---|
| Base (LLM-only) | 43.3% | 9 | 1450 | 180 | 2100 | 0.018 | – |
| Examples | 53.3% | 8 | 2400 (↑1.66x) | 210 (↑1.17x) | 3200 (↑1.52x) | 0.036 (↑2.00x) | 5.56 |
| Annotations | 56.7% | 8 | 2200 (↑1.52x) | 190 (↑1.06x) | 3000 (↑1.43x) | 0.032 (↑1.78x) | 9.57 |
| Validation | 46.7% | 5 | 1500 (↑1.03x) | 240 (↑1.33x) | 3400 (↑1.62x) | 0.025 (↑1.39x) | 4.86 |
| Examples + Validation | 46.7% | 5 | 2500 (↑1.72x) | 260 (↑1.44x) | 4300 (↑2.05x) | 0.041 (↑2.28x) | 1.48 |
| Annotations + Validation | 66.7% | 4 | 2800 (↑1.93x) | 240 (↑1.33x) | 4200 (↑2.00x) | 0.045 (↑2.50x) | 8.67 |
| Examples + Annotations | 83.3% | 2 | 3600 (↑2.48x) | 250 (↑1.39x) | 5200 (↑2.48x) | 0.058 (↑3.22x) | 10.00 |
| Examples + Annotations + Validation | 90.0% | 0 | 3900 (↑2.69x) | 290 (↑1.61x) | 6100 (↑2.90x) | 0.067 (↑3.72x) | 9.53 |

**Retrieval Quality Remains a Core Bottleneck.** Retrieval quality strongly affects downstream SQL generation. A substantial gap remains between Gold-table access and retrieval-based settings even with advanced prompting and validation. Increasing retrieval depth improves schema recall but also introduces more noise, larger prompts, higher token consumption, and increased latency.

**Accuracy-Performance-Cost Tradeoffs.** Higher-performing configurations generally require larger prompts, more tokens, and greater inference costs. Examples increase token consumption, annotations lengthen prompts, and validation adds latency through regeneration cycles. As a result, the most accurate configurations are not always the most operationally efficient, highlighting the need to evaluate text-to-SQL systems beyond execution accuracy alone.

**Implications for Enterprise Text-to-SQL.** The results suggest that effective enterprise text-to-SQL systems require joint optimization of retrieval, prompting, validation, and efficiency. The strongest configurations combine semantic schema grounding, structural guidance, and robustness mechanisms, but their effectiveness depends heavily on retrieval quality and schema complexity. Future systems may benefit from adaptive architectures that dynamically balance these components according to query difficulty and operational constraints.

## 7 CONCLUSION

This paper presented a real-world enterprise text-to-SQL use case from industry, a modular pipeline we built to handle this use case, and an ablation study for evaluating the tradeoffs across a variety of pipeline configurations as well as accuracy-performance-cost metrics. We evaluated the impact of several prompt augmentation strategies, including few-shot examples, schema annotations, and iterative SQL validation across multiple schema exposure settings. In addition to execution accuracy, the evaluation analyzed robustness, execution failures, latency, token consumption, and monetary inference cost. Overall, our study demonstrates that enterprise text-to-SQL optimization is fundamentally a multi-objective problem involving correctness, robustness, retrieval quality, latency, and inference cost. Future work includes exploring adaptive retrieval strategies, learned schema linking, multi-stage reasoning pipelines, and agentic text-to-SQL architectures capable of dynamically balancing accuracy and deployment efficiency. In addition, future studies should evaluate the proposed pipeline on larger benchmark sets

to improve the statistical robustness of the conclusions and investigate whether the observed trends generalize to smaller open-source language models.

## ACKNOWLEDGMENTS

This research is supported by Intel as part of the MIT Data Systems and AI Lab (DSAIL). Omar Bouattour was supported by the Konrad Zuse School of Excellence in Reliable AI (relAI). The authors also thank all collaborators and colleagues who contributed valuable feedback and discussions throughout this work, including Moh Haghighat and Swaathi Sampath Kumar from Intel for their input on the dataset. OpenAI's ChatGPT was used to assist in generating Figure 1, improving the manuscript text, and refining portions of the code. All generated content was reviewed, verified, and edited by the authors, who take full responsibility for the final manuscript.

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
