# OpenReview forum: "Prompting Strategies for Enterprise Text-to-SQL: An Ablation Study"
_VLDB.org/2026/Workshop/NOVAS — NOVAS 2026_

### Official Review · Reviewer_UEqc · 2026-06-29

**Confidence:** 3

**Improvement Opportunities:**

O1. The results are based on only 30 queries. An outlier behavior from any query can swing the accuracy results significantly. For future full-length publication, more testing queries are expected to draw robust statistical conclusions.

O2. I’m curious how the conclusions change if the experiments are carried out with smaller open-source models instead of GPT4.1.

O3. The paper concludes the evaluation section with the statement “future systems should move towards adaptive strategies capable of dynamically balancing accuracy and efficiency,
depending on query complexity and operational constraints.” It would be nice to include a brief discussion about the possible ways to achieve such adaptivity.

**Minor Comments:**

M1. Some of the text in Figure 1 is too small.

**Short Summary:**

This paper presents an empirical ablation study of a modular Text-to-SQL pipeline. The authors systematically evaluate the interplay between three prompt augmentation components: few-shot examples, schema annotations, and iterative SQL validation. The authors find that semantic schema annotations provide the highest performance gains. However, achieving a high accuracy via a combination of all techniques introduces significant operational overhead.

**Strong Points:**

S1. The study is performed with a highly practical dataset (i.e., the Intel telemetry dataset) with cryptic terminology and wide variety of information.

S2. The paper evaluates  operational metrics alongside accuracy, framing Text-to-SQL as a multi-objective optimization problem.

S3. The finding that few-shot examples can actually degrade performance when schema retrieval introduces noise is interesting.

S4. The discussion section summarizes the valuable lessons from the emprical study well.

---

### Official Review · Reviewer_VmoU · 2026-07-08

**Confidence:** 4

**Improvement Opportunities:**

Thank you for submitting to NOVAS. I liked your work. Please find my comments below.

O1. Is it possible to make Intel's telemetry dataset public? It would be immensely useful for the community. Microsoft did that at some point for their Azure telemetry data [1], and it was very useful for research. They required signing a document and sending it to them, and they would provide the data on an SSD.

[1] https://dl.acm.org/doi/10.1145/2806777.2806845

O2. Similarly, is it possible to open your benchmark to compare with existing public benchmarks? You do compare your benchmark with the public benchmarks (section 3), but if we had the queries somewhere as supplementary material, for example, that would make the paper much stronger and also would be really useful for the community. Perhaps, if not for the current paper, it could be part of a follow-up paper. Real-life benchmarks are really useful for us to ideate on new system architectures and solutions.

O3. I think you are using an LLM for SQL validation. What is that so? Can't we validate SQL with a compiler? SQL has a predefined syntax, and it should be relatively easy to detect whether a given SQL is syntactically accurate or not, isn't it? Why use an LLM for that and risk multiple trials and achieve a negative result? In the experimental results, there were quite many syntactic errors (orange bars), which despite that you run the validation multiple times ("This process may repeat for a predefined number of retries.").

O4. Following up, how many times do you repeat the SQL validation loop on average? Does it take a lot of repetitions? And, how many times did you actually reach the limit?

O5. In the evaluation metrics, you mention you tolerate numeric mismatches. How aggressively are you doing it? A numeric imprecision could as well be interpreted as an approximate result, isn't it?

O6. "This setting isolates SQL generation performance from schema retrieval quality and approximates an oracle upper bound." --> I think this sentence is a bit misleading. It doesn't really give an upper bound. This is one of the optimization dimensions, like schema enrichment. You simply analyze it step by step, which makes a lot of sense. But it isn't really giving an upper bound.

O7. Back to my validation comment (O3), Figure 3 seems to imply that if we had eliminated validation errors with a rule-based solution, e.g., a compiler, we could achieve a whole different set of results, can't we:

"Validation becomes more important under retrieval noise, because retrieval errors increase the likelihood of malformed SQL or invalid
references. Unexpectedly, the full configuration slightly underperforms Validation with Annotations."

O8. Table 1 is great. So, there is one explanation for all these results: cost. It doesn't come for free. The more information we add, the better results we get. Can we perhaps compare different optimizations within the same budget? Could we have such a metric of accuracy per-token, and perhaps see which optimization is the best one among those?

O9.

**Minor Comments:**

N/A.

**Short Summary:**

The paper presents an LLM-based Text-to-SQL system analysis based on real-life Intel telemetry data. The results are interesting, and the analysis thorough.

**Strong Points:**

S1. Important problem.

S2. Interesting methodology.

S3. Interesting results.

---

### Official Review · Reviewer_275z · 2026-07-11

**Confidence:** 4

**Improvement Opportunities:**

O1. The main feedback I have is regarding the novelty of the conclusions and work. It has been a constant in Text-to-SQL analysis to rely on some form of ablation to understand the benefits of various components proposed and the majority of Text-to-SQL pipelines in the literature are modular by design (i.e., multi-stage). Very few of the conclusions seem actionable or different (retrieval being the hardest is in part because not having a 100% recall means you cannot obtain a correct query). While I enjoyed seeing that cost is part of the analysis, I would like to note that there has been prior work on LLM model routing as a way of reducing cost applied to many problems including Text-to-SQL.

If you consider for instance the following paper: "DIN-SQL: Decomposed In-Context Learning of Text-to-SQL with Self-Correction. NeurIps 2023". It contains an ablation comparing the use or not of various forms of self-correction (equivalent to validation), impact of schema linking and the use of few-shot and CoT prompting. Many prior Text-to-SQL work compares their techniques when providing the ground truth schema and while doing retrieval across various implementations. Granted many evaluations were done on Spider and Bird as benchmarks but the conclusions are not different.

O2. Evaluation analysis can be improved and I will point a couple of aspects in the few-shot example part:

2.1 One of the recommendations regarding few-shot examples was: "they should be used selectively when retrieval quality and context budgets permit". I understand having budget as a constraint but I doubt that it is possible to evaluate the retrieval quality for a query of which we presume no access to ground-truth schema in production. It wasn't clear how to take the retrieval quality into account and make this actionable.

2.2 The evaluation (Section 5.2) mentions that "While Examples can improve structural reasoning, they also increase latency and inference cost considerably." While I agree that there would be an increase in cost with the caveat that output tokens are 6x more expensive than input ones, I am not sure if the latency claims are as conclusive. From the experiments, the best solution increases the output tokens by 1.6x compared to the baseline and they have a lot more impact on latency than the input tokens due to the autoregressive generation. The numbers would also vary a lot due to the use of a private cloud with no control over the deployment.

2.3 It is probably best to report total and median metrics for tokens and cost.

**Minor Comments:**

M1. Reporting evaluation analysis with an old model might lead to a general bias of future readers against the results even though they are perfectly valid and unlikely to change. Note that I do understand the associated costs of rerunning these experiments in an academic setting.

**Short Summary:**

The authors ran an ablation study of prompting techniques using a private set of queries from an Intel telemetry use case. The goal of the work is to share takeaways and provide guidelines for practitioners regarding the use of prompt augmentation, schema annotations, few-shot examples, and validation while taking into account the operational cost.

**Strong Points:**

S1. Systematic experimental analysis that is well designed as part of the ablation.

S2. The evaluation uses real queries from production with their natural language counter part annotated by domain experts.

S3. It is uncommon for Text-to-SQL evaluation papers to also report cost unless that's the target, which is important for handling a large volume of queries for an actual deployment.

---

### Decision · Program_Chairs · 2026-07-16

**Decision:**

Accept

**Comment:**

This paper presents a systematic ablation study of prompt augmentation, schema annotations, and SQL validation using real Intel telemetry queries. Its production-oriented dataset, careful methodology, and joint analysis of accuracy and operational cost provide valuable guidance for practical Text-to-SQL deployments. The findings on schema annotations, retrieval noise, and few-shot examples are of high interest for the NOVAS community.